# Beyond mono fertilization: Mixed fertilization enhances productivity and quality of chili (*Capsicum frutescens*)

Hossain Md Dalim[1⊕], Md Golam Jilani Helal[1⊕], Minhazul Kashem Chowdhury[2], Shoaib Rahman[3], Sohel Rana Mazumder[2], Md. Ismail Hossain[4], Md. Mohi Uddin Sujan Chowdury[1], Md. Shahariar Jaman[1*]

1 Department of Agroforestry and Environmental Science, Ecosystem Ecology Lab, Sher-e-Bangla Agricultural University, Dhaka, Bangladesh, 2 Department of Horticulture, Sher-e-Bangla Agricultural University, Dhaka, Bangladesh, 3 Department of Plant Pathology, Sher-e-Bangla Agricultural University, Dhaka, Bangladesh, 4 Department of Agronomy, Sher-e-Bangla Agricultural University, Dhaka, Bangladesh

⊕ These authors contributed equally to this work.
* shmishu2287@sau.edu.bd

## Abstract

While the effects of different fertilization strategies on chili (*Capsicum frutescens*) cultivation have been previously examined, the lack of comparative assessment of mixed versus mono-fertilization approaches limits our understanding. To address this gap, we conducted an experiment using a completely randomized design (CRD) with five fertilization treatments along with four replications of each treatment (e.g., *Leucaena leucocephala* leaf litter and recommended dose of synthetic fertilizers (RDF)) to observe the effect of these treatments on productivity traits, NPP (net primary productivity), quality, and overall yield of chili. We found that mixed fertilization ($T_4$) results higher yield (~ 60% and ~ 90%) compared to control ($T_0$) and sole organic fertilizer ($T_1 + T_2$) respectively. The $T_4$ treatment showed consistently higher plant height, leaf and fruit number at 60 DAT, as well as higher flowering at 45 DAT compared to other treatments. We also found that NPP (above- and belowground combined) was higher in $T_3$ and $T_4$ treatments compared to control and organic fertilization. Similarly, quality traits such as vitamin C, capsaicin content, and SPAD reading were higher under mixed fertilization. Linear fit regression model indicated that both ANPP and yield were positively associated with vegetative and reproductive traits, particularly leaf number and flowering, highlighting that structural growth directly contributed to productivity gains. Overall, our results suggest that mixed fertilization enhances both productivity and quality traits of chili. Therefore, integrated organic and inorganic fertilizer management are recommended to improve yield and quality of chili.

**Data availability statement:** The relevant data and R script used for the data analysis are included as supplementary information.

**Funding:** Our research was funded by the Sher-e-Bangla Agricultural University Research System (SAURES) grant no. SAU/SAURES/2022/110(20) and Ministry of Science and Technology, Bangladesh, through a special research allocation grant (Grant No: SRG-221311(BS)/2022-23), however, the funder had no role in this study regarding design, data compilation, analysis, any decision or publication of this manuscript. Additionally, we acknowledge our funder in the acknowledgement section of our manuscript.

**Competing interests:** The authors declare that they have no competing interests.

# 1. Introduction

Chili (*Capsicum frutescens*) is a widely used spice that belongs to the *Capsicum* genus of the *Solanaceae* family [1] and is now extensively cultivated around the world [2]. It is indigenous to America and the West Indies, but began its global spread to tropical regions after those areas were explored [3]. Bangladesh is no exception, as chili cultivation occupies a significant portion of the country's agricultural landscape. For instance, during the 2022–2023 chili was cultivated on approximately 188,888.78 ha of land in Bangladesh and produced about 507,200.88 metric tons of yield [4]. As global consumption of chili increases, the demand for improved fruit quality and overall productivity has increased [5]. Therefore, meeting this demand depends heavily on effective nutrient management.

Successful chili production primarily depends on fertilizers, and farmers often rely on mineral fertilizers for rapid visible results. Inorganic inputs such as urea and NPK blends supply nutrients in forms readily available to plants, which accelerates vegetative growth and fruiting [6,7]. However, these advantages also bring some problems. Continuous or excessive use of inorganic fertilizers may cause nutrient imbalances, soil acidification, losses in microbial diversity, and reduce soil fertility over time [8].

In contrast, organic fertilizers contribute in ways that go beyond immediate crop growth. Composts, manures, and green residues enhance soil structure, build organic matter, and foster beneficial microbial activity [9,10]. These qualities improve long-term soil health, water retention, and nutrient cycling. Yet, the release of nutrients from organic sources is often slower and less predictable compared with inorganic fertilizers. This creates a challenge for high-yielding chili varieties that require both timely nutrient availability and sustained fertility. Thus, neither organic nor inorganic fertilization alone is likely to provide a complete solution.

A strategy that combines both organic and inorganic fertilizers is needed to balance productivity with sustainability. Mixed fertilization strategies, which integrate both organic and inorganic nutrient sources, have been recognized as a holistic and sustainable approach for soil fertility and crop management. These integrated nutrient strategies aim to exploit the rapid nutrient availability of inorganic fertilizers, while simultaneously harnessing the long-term soil health provided by organic amendments such as green manures and composts [11]. Specifically, the incorporation of *L. leucocephala* leaf litter along with inorganic fertilizers has shown potential to enhance crop productivity and soil health by synchronizing nutrient release with crop demand, reducing reliance on synthetic inputs, and promoting ecological sustainability [12,13]. These combined effects may contribute to a balanced nutrient environment, supporting plant metabolism, soil biological activity, and overall system resilience [14]. For instance, the combined application of inorganic fertilizers with organic sources has been reported to improve vegetative traits, nutrient uptake, and fruit quality of *Capsicum* species, such as vitamin C, capsaicin, and chlorophyll [15–17]. While these findings demonstrate the potential of integrated approaches, mostly focusing on crop responses, the comprehensive investigations specifically targeting species like *Capsicum frutescens* remain limited. In particular, the influence of mixed fertilization

on net primary productivity (NPP), fruit quality traits, and yield-contributing characters of chili has not been sufficiently examined.

To address this gap, the present study investigates a comparative performance of mono versus mixed fertilization on productivity and quality traits of chili. Thus, this research aims to clarify whether mixed fertilization provides measurable advantages over sole organic or inorganic inputs. Therefore, we hypothesized that integrating *L. leucocephala* leaf litter with inorganic fertilizers would outperform sole applications of either organic or inorganic fertilizer on chili cultivation. Our specific objectives are (i) to analyze the effect of mixed and mono fertilization on potential yield contributing traits of chili; (ii) to find out the effect of mixed and mono fertilization on NPP and quality of chili, and (iii) to evaluate the relationships between potential yield contributing traits with ANPP and yield of chili.

## 2. Materials and methods

### 2.1. Experimental site and plant material

The experiment was conducted outdoors from July to December 2023 at the agroforestry field laboratory of the Department of Agroforestry and Environmental Science, Sher-e-Bangla Agricultural University, under open-environment pot conditions. The site is geographically positioned at 23.77°N latitude and 90.35°E longitude, with an elevation of approximately 8.6 meters above sea level. The region experiences a subtropical monsoon climate with significant rainfall during the monsoon and moderate temperatures year-round. During the experimental period, the average monthly temperature ranged from 23.8 °C to 31.4 °C, while the average relative humidity fluctuated between 67% and 87%. The total rainfall recorded during the experimental period was approximately 1100 mm (range from 20 to 370 mm), with the highest precipitation occurring in July and August (Fig 1) and the sunshine hours varied between 4.3 and 7.2 hours per day [18].

The experimental soil was classified as silty loam with good drainage and moderate fertility. Composite soil samples were collected before the experiment. The plant material used was chili (*Capsicum frutescens*), variety BARI-3, obtained from the Bangladesh Agricultural Research Institute (BARI). This variety is well known for its adaptability, pest resistance, and high yield potential. Seedlings were raised in a nursery and transplanted into pots just after 20 days of sowing, with an average seedling height of 15 cm at transplanting.

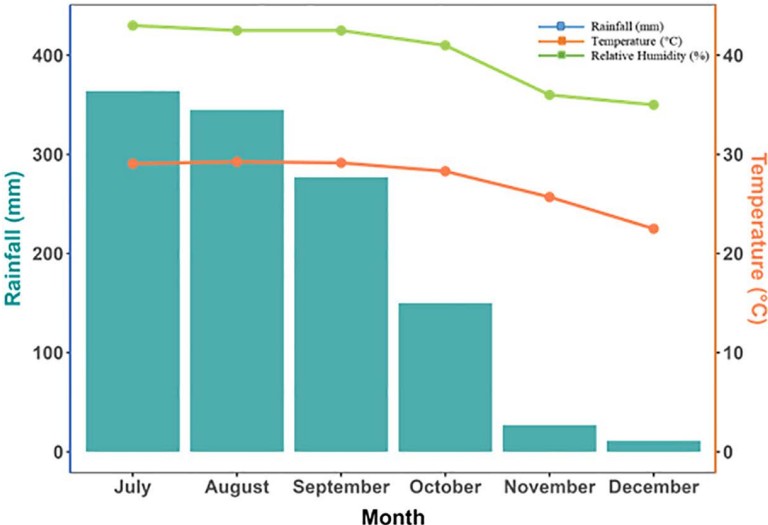

**Fig 1. Meteorological data (rainfall (mm), temperature (°C), and relative humidity (%)) during the experiment period (July–December 2023) were derived from BMD (Bangladesh Meteorological Department), Sher-e-Bangla Nagar, Dhaka.**

## 2.2. Pot preparation and experimental design

The experiment was laid out in a completely randomized design (CRD) comprising five fertilization treatments, and each treatment was replicated four times to obtain 20 pots for seedling transplantation (Fig 2). The design is suitable for random allocation of treatments, bias minimization, and maintain controlled along with relatively uniform experimental conditions. Modern nursery plastic pots (made of Polypropylene) measuring 30 cm in height and 28 cm in diameter (approximately 18–20 liters capacity) were used. Pots are designed and produced by Rangpur Foundry Limited (RFL plastic company), particularly for seedling and sapling raising. Each pot was thoroughly cleaned, perforated at the base for drainage, and a gravel layer was added to prevent waterlogging. Topsoil from the upper 0–20 cm layer was collected, air-dried, sieved (2 mm mesh), and homogenized to ensure uniform texture and fertility. Each pot was filled with 7.5 kg of prepared soil.

Inorganic fertilizers (urea, triple superphosphate (TSP), muriate of potash (MOP), gypsum, boric acid, and zinc sulfate), and organic material (oven-dried *L. leucocephala* leaf litter) were incorporated into the soil one week before transplanting according to treatment specifications. After treatment incorporation, the soil was stabilized, and two healthy 20-day-old chili seedlings (collected from Bangladesh Agricultural Research Institute- BARI-3 variety) were transplanted. Later, chili seedlings in each were monitored for early growth performance. After establishment, one seedling was carefully removed from each pot, allowing a single plant to grow until maturity. This adjustment was made because the pot size (30 cm height × 28 cm diameter) was not suitable for maintaining two seedlings, which could create above- and below-ground competition and confound treatment effects. Maintaining one seedling per pot ensured uniform growth and accurate assessment of nutrient utilization under each treatment. Pots were arranged randomly with 1.0 m spacing to ensure

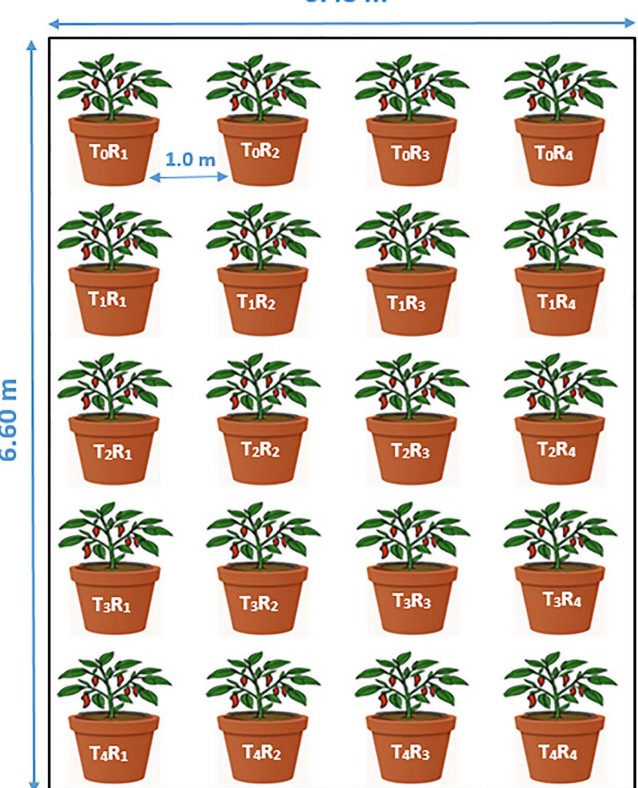

**Fig 2. Field lay out of the experimental plot.**

uniform sunlight exposure and airflow. All pots were labeled clearly, and crop management practices were applied uniformly across treatments throughout the experiment. Standard agronomic practices were uniformly applied across all pots. Regular watering was performed to maintain field capacity, and manual weeding was done as needed. Pest and disease control measures were implemented using recommended safe practices to ensure uniform crop health throughout the growing season.

Before treatment application, composite samples were analyzed for pH and key nutrient contents. Parallel nutrient analysis was carried out for the *L. leucocephala* leaf litter intended for organic amendments. Incorporation of *L. leucocephala* leaf litter added 1.19% nitrogen, 0.16, 0.15, 2.43 mg kg$^{-1}$ phosphorus, sulfur, and boron respectively. Additionally, *L. leucocephala* leaf litter provides 0.04 and 0.48 mEq 100g$^{-1}$ potassium and magnesium in our experimental soil (Table 1).

**2.2.1. Fertilizer application.** Fertilizer treatments combined with organic and/or inorganic nutrient sources in different proportions are shown in Table 2. The inorganic fertilizers, e.g., urea, TSP, MOP, gypsum, boric acid, and zinc sulfate, were applied either at the full recommended dose (RDF) or at half the RDF, calculated according to national agronomic guidelines and converted to pot$^{-1}$ quantities. Each inorganic fertilizer played a specific role in chili growth, such as urea, which provides nitrogen that supports vegetative growth and chlorophyll formation, where TSP and MOP provide phosphorus and potassium respectively, which enhance root growth, fruit set, water regulation, sugar transport, and fruit quality [16,17,19–21]. In addition, gypsum provides calcium and sulfur that strengthen cell walls and promote amino acid synthesis, while boric acid and zinc sulfate provide boron and zinc essential for pollen germination, fruit formation, chlorophyll synthesis, and enzyme activity [11,16,22–24].

**Table 1. Nutrient composition of the experimental soil before treatment application and nutrient composition of the decomposed *L. leucocephala* leaf litter.**

| Nutrient parameter | Unit | *L. leucocephala* leaf litter | Initial soil before experiment |
|---|---|---|---|
| pH | — | 6.9 | 6.5 |
| Total Nitrogen (N) | % | 1.19 | 0.08 |
| Phosphorus (P) | mg/kg soil | 0.16 | 92.72 |
| Potassium (K) | mEq/100g | 0.04 | 0.66 |
| Sulfur (S) | mg/kg soil | 0.15 | 46.14 |
| Magnesium (Mg) | % (leaf)/ mEq/100g (soil) | 0.48 | 9.48 |
| Boron (B) | mg/kg soil | 2.43 | 0.78 |

**Table 2. Rates of different fertilizer doses varied by treatments.**

| Treatment | Organic fertilizer | Inorganic fertilizer* |
|---|---|---|
| $T_0$ | None | RDF |
| $T_1$ | L. *leucocephala* leaf litter- 12.4 g kg$^{-1}$ soil (93 g pot$^{-1}$) | None |
| $T_2$ | L. *leucocephala* leaf litter- 24.8 g kg$^{-1}$ soil (186 g pot$^{-1}$) | None |
| $T_3$ | L. *leucocephala* leaf litter- 12.4 g kg$^{-1}$ soil (93 g pot$^{-1}$) | ½ RDF |
| $T_4$ | L. *leucocephala* leaf litter- 24.8 g kg$^{-1}$ soil (186 g pot$^{-1}$) | ½ RDF |

*RDF- Urea (1.0 g), TSP (1.10 g), MOP (0.90 g), Gypsum (0.60 g), Boric acid (0.25 g), and zinc sulfate (0.25 g). ½ RDF- Urea (0.50 g), TSP (0.55 g), MOP (0.45 g), Gypsum (0.30 g), Boric acid (0.12.5 g), and zinc sulfate (0.12.5 g) [Fertilizer application rates were calculated and converted to g pot$^{-1}$ according to RDF regulation of t ha$^{-1}$] [25–27].

**2.2.2. Application of litterbag technique.** To investigate the decomposition dynamics and nutrient release patterns of *L. leucocephala* leaf litter, the **litterbag technique** was employed, a widely accepted method for studying organic matter breakdown in agroecology [28,29]. We followed the negative exponential decay model to assess decomposition dynamics of *L. leucocephala* leaf litter [30]

$$k = -Ln\ (X_t/X_o)/t$$

Where, $X_0$ = initial dry weight, $X_t$ = dry weight at time t, k = decomposition constant, and t = time (days).

We used nylon mesh bags (2 mm) containing 10 g of air-dried litter buried 5 cm deep in relevant treatment pots to allow microbial access while retaining material integrity [31]. Litterbags were retrieved 4 times, e.g., at 15-day intervals over 60 days after placement. After collection, the litter was gently washed to remove adhering soil particles, then oven-dried at 65 °C until a constant weight was reached and subsequently weighed to determine mass loss as an indicator of decomposition and nutrient mineralization [32]. Our experimental *L. leucocephala* leaves exhibited faster decomposition with a decay constant (k) and exhibited approximately ~70% mass loss within 60 days, indicating fast nutrient mineralization. Later, we deployed only decomposed litter to our experimental pot soil.

## 2.3. Data collection

In this study, various growth, yield, productivity, and quality parameters of chili (*Capsicum frutescens*, var. BARI-3) were measured following standardized data handling procedures. Details for each parameter are provided below:

Plant growth was monitored through regular measurements of plant height and leaf count at 15, 30, 45, and 60 days after transplanting. Plant height was measured from the soil surface to the apical tip of the main stem using a standard ruler. Fully developed leaves were counted manually for each plant, excluding senescent and immature leaves. Flowering and fruiting were recorded at 30 and 45 days after transplanting, with a final fruit count taken at 60 days to monitor full crop development. All fully opened flowers and mature fruits were counted manually per plant and averaged across replications. Aboveground Net Primary Productivity (ANPP) was assessed at maturity by harvesting all shoot biomass (leaves and stems), which was oven-dried at 60 °C for 48 hours to constant weight and expressed in g plant$^{-1}$ [33]. Belowground Net Primary Productivity (BNPP) was determined by carefully uprooting plants and placing them in plastic bags, stored at 4 °C until processing. Sampling was done up to a 10 cm soil depth to recover most root biomass. Roots were separated from soil by gently washing under running water through a 2 mm mesh sieve to collect both fine and coarse roots. Cleaned roots were oven-dried at 60 °C for 48 hours or until a constant weight was reached. After drying, samples were cooled in a desiccator and handled with clean, dry forceps. The dried roots were then transferred into labeled brown paper envelopes. Final root biomass was measured using a precision digital balance and recorded as dry root biomass in g plant$^{-1}$ [33]. Fruit yield per plant was recorded at final harvest using a precision digital balance (HOCHOICE, Model no: HC20001X). The yield (t ha$^{-1}$) was extrapolated using a formula derived from a brief guide to estimating horticultural crop yields (www.agriculture.vic.gov.au). Furthermore, we cross-checked our yield estimation according to the formula given by Gomez & Gomez [34] and the guideline of estimating the yield of horticultural crops given by Bangladesh Agricultural Research Institute (www.bari.gov.bd).

$$\text{Yield (t/ha)} = \frac{\text{Yield per pot (kg)} \times 10000}{\text{Pot area (m}^2) \times 1000}$$

Vitamin C content was quantified using the 2, 6-dichloroindophenol titration method [35]. Firstly, the samples were heat-treated at 70 °C and dried until a constant weight. Later, the samples were powdered and titrated to determine ascorbic acid concentration, expressed as mg 100 g$^{-1}$ fresh weight. Capsaicin content was determined colorimetrically using a Thermo GENESYS 10 UV spectrophotometer. Absorbance was measured at 286 nm and compared against a standard

capsaicin calibration curve (0–0.10 mg mL$^{-1}$) prepared in a methanol: ethanol: water (6:2:2 v/v) solution [36]. The results were expressed as mg capsaicin g$^{-1}$ dry sample. Chlorophyll content was estimated using a SPAD meter (SPAD-502; Konica Minolta Sensing, Inc., Osaka, Japan). For each plant, three leaves were randomly selected, and SPAD readings taken at their midpoints were averaged and obtained accuracy.

## 2.4. Data analysis

All collected data were subjected to rigorous statistical analysis to evaluate the effects of mixed and mono fertilization on the growth, productivity, and quality traits of *Capsicum frutescens*. The analysis was performed using R software (version 4.3.1; R Core Team, 2023) [37], and figures were generated using the ggplot2, tibble, and plyr packages [38]. Additional R packages, such as ggpubr [39], ggpmisc [40], gridExtra [41] and nlme [42], were also utilized for graph plotting. Prior to hypothesis testing, data were assessed for normality and homogeneity of variance using the Shapiro-Wilk test [43], and Levene's test [44], respectively to ensure compliance with the assumptions of parametric analysis. Variables that met these assumptions were subjected to one-way analysis of variance (ANOVA) to determine the significance of treatment effects across all measured parameters, including morphological (plant height, leaf number), reproductive (flower and fruit number), biomass productivity (ANPP, BNPP), yield and fruit quality traits (vitamin C content, capsaicin concentration and SPAD reading). Where ANOVA indicated significant differences, treatment differences and average values were partitioned with Tukey's Honestly Significant Difference (HSD) test, where the differences were predicted at $p < 0.05$ significance level [45]. Next, linear fit ($R^2$) regression analysis was performed with the 'lm' function to see the bivariate relation (e.g., number of leaves per plant, number of flowerings per plant, with ANPP, and chili yield) and significance was indicated at $p < 0.05$, $p < 0.01$, and $p < 0.001$ levels. We used F-statistics to determine whether the variance between two standard variables is similar.

## 3. Results

### 3.1. Mixed fertilization enhances vegetative and reproductive traits of chili across different growth stages

Plant height increased significantly under $T_0$ treatment at 60 DAT ($58.25 \pm 5.96$ cm; $p = 0.003$) compared with 15 DAT ($28.75 \pm 1.38$ cm), where no significant variation ($p > 0.05$) was found in $T_1$ and $T_2$ treatments among all observations at 15, 30, 45, and 60 DAT respectively. In contrast, $T_3$ and $T_4$ treatments at 45 DAT ($T_3$: $56.50 \pm 4.99$ cm, $p < 0.001$; $T_4$: $66.75 \pm 7.60$ cm, $p = 0.007$) and 60 DAT ($T_3$: $63.25 \pm 4.64$ cm, $p < 0.001$; $T_4$: $73.75 \pm 7.87$ cm, $p = 0.007$) showed significant treatment differences compared to 15 DAT (T3: $33.00 \pm 1.47$ cm; T4: $35.00 \pm 3.03$ cm) (Fig 3a). On in one hand, $T_0$ treatment observed the higher number of leaves ($824.25 \pm 50.13$; $p < 0.001$) at 60 DAT, compared to 15 DAT ($198.25 \pm 22.26$) and 30 DAT ($46 8.50 \pm 36.27$) leaves plant$^{-1}$, similarly, $T_1$, $T_2$, $T_3$ and $T_4$ treatments showed the significant ($p < 0.001$) differences among 60, 30 and 15 DAT respectively (Fig 3b). Except $T_0$ and $T_4$ treatments for flowering, $T_1$, $T_2$ and $T_3$ treatments showed significant treatment variation ($p < 0.05$) at 30 and 45 DAT respectively (Fig 3c). For fruit number, $T_0$ and $T_1$ treatments observed no significant differences ($p > 0.05$) at 30, 45 and 60 DAT but $T_2$, $T_3$ and $T_4$ treatments exhibited the significant differences ($p < 0.05$) among all the observations, while T4 showed the highest result at 60 DAT ($93.00 \pm 5.49$ fruits plant$^{-1}$). When we averaged organic ($T_1 + T_2$) and mixed fertilizer treatments ($T_3 + T_4$), we found mixed fertilizer ($T_3 + T_4$) showed significantly higher plant height, leaf number, flowering, and fruit set at all DAT ($p < 0.05$) compared to control ($T_0$) and organic fertilizers ($T_1 + T_2$) respectively.

### 3.2. Net primary productivity maximized under mixed inputs

For ANPP and BNPP, there were no significant treatment differences ($p > 0.05$) among $T_0$, $T_3$, and $T_4$ treatments. When we averaged organic ($T_1 + T_2$) and mixed fertilizer treatments ($T_3 + T_4$), we found ANPP in mixed fertilization treatment ($T_3 + T_4$) showed significant differences ($21.82 \pm 2.20$ g plant$^{-1}$; $p < 0.001$) compared to other treatments, which representing 26.67%

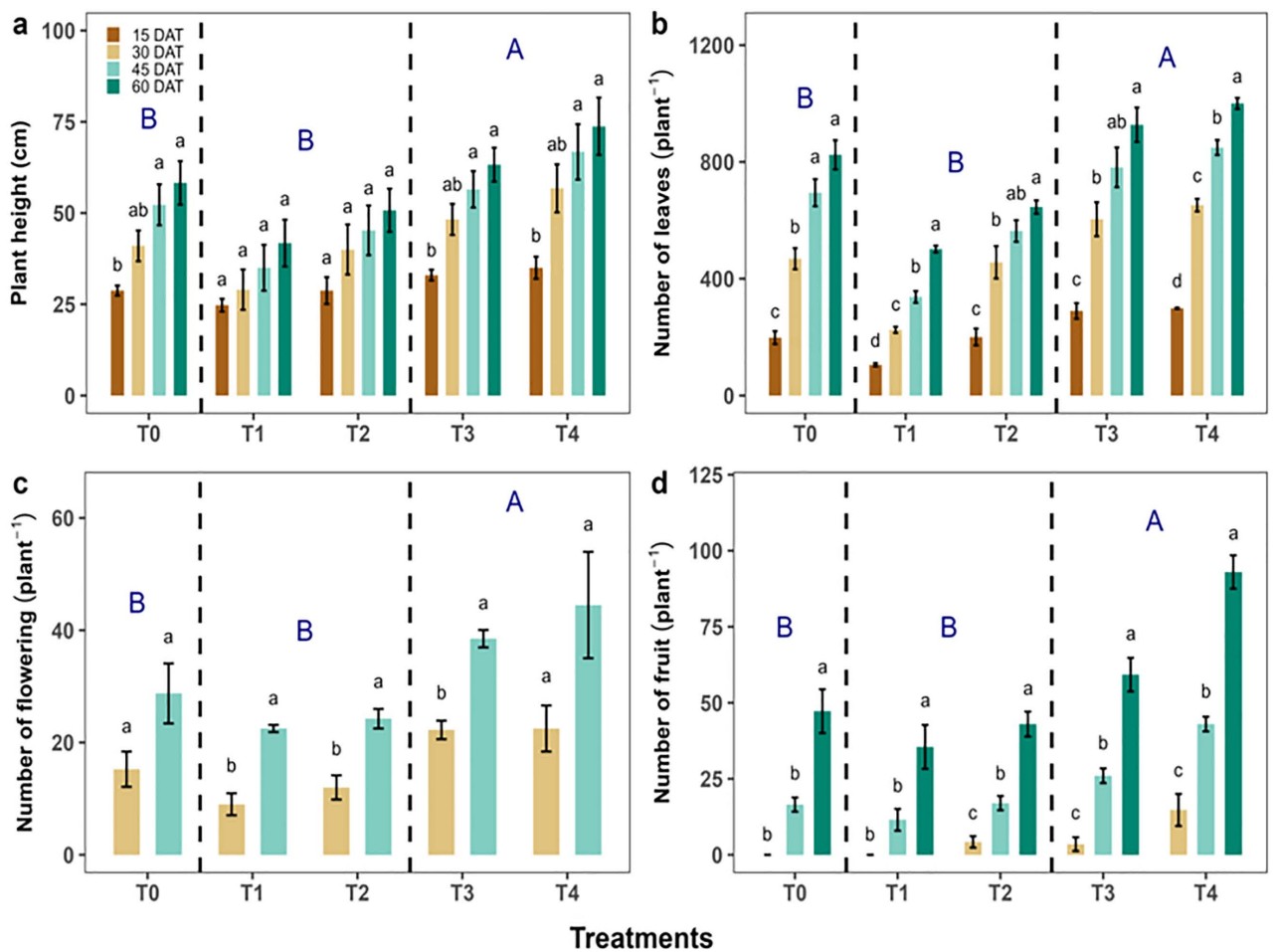

**Fig 3. Effect of different fertilization treatments on plant height, number of leaves, flowering, and fruit production of *Capsicum frutescens* at various DAT.**

and 116.39% increase compared to inorganic ($T_0$) and organic ($T_1 + T_2$) fertilizer respectively (Fig 4a). Root biomass (BNPP) enhanced by mixed fertilization reaching ($3.93 \pm 0.39$ g plant$^{-1}$) under ($T_3 + T_4$) treatments, which was significantly greater than mono fertilization ($p < 0.001$) (Fig 4b). Total NPP (ANPP + BNPP) was found higher in mixed fertilization ($T_3 + T_4$) treatments ($26.47 \pm 3.16$ g plant$^{-1}$) followed by organic ($T_1 + T_2$) ($13.84 \pm 1.37$ g plant$^{-1}$) and control ($20.48 \pm 2.47$ g plant$^{-1}$; $p = 0.01682$) respectively (Fig 4c).

### 3.3. Mixed fertilization enhanced fruit yield

Mixed fertilizer produced ($13.01 \pm 1.24$ t ha$^{-1}$; $p = 0.003$) fruit yield t ha$^{-1}$, which was 37.48% and 60.29% higher than the inorganic ($T_0$) organic fertilizer ($T_2 + T_4$) treatments respectively (Fig 5). At the same time, $T_4$ treatment produced ($15.50 \pm 0.54$ t ha$^{-1}$) fruit yield, which was 63.90% higher than the control, as well as 106.21% above $T_1$ and 78.02% over $T_2$ treatments. In contrast, the $T_3$ treatment ($10.51 \pm 1.92$; $p = 0.004$ t ha$^{-1}$) showed a significant difference from the $T_4$ treatment but no significant variation compared with $T_2$ ($8.70 \pm 1.57$), $T_1$ ($7.52 \pm 0.81$), and $T_0$ ($9.46 \pm 0.76$) yield t ha$^{-1}$ treatments respectively (Fig 5).

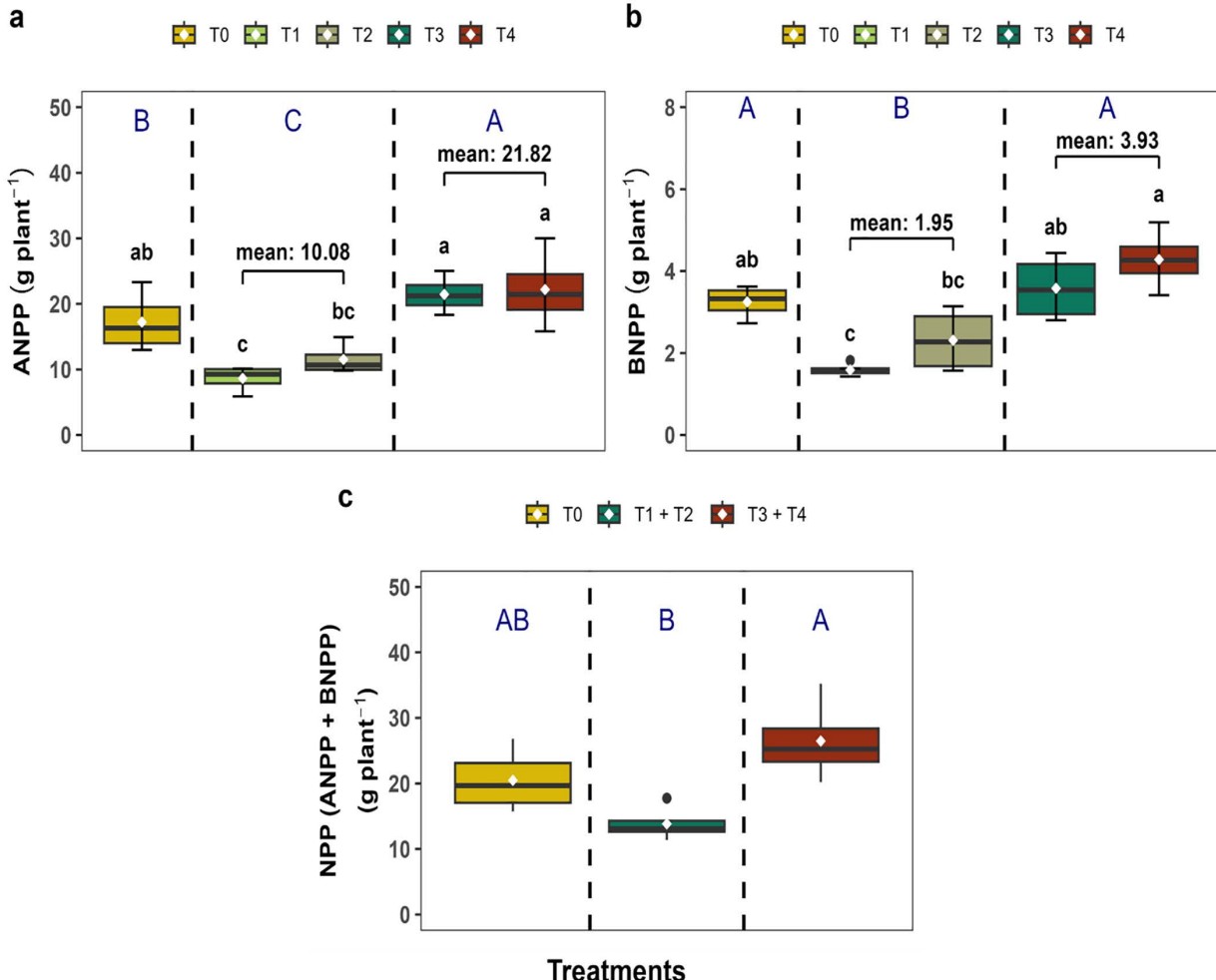

**Fig 4.** **(a) Aboveground Net Primary Productivity (ANPP), (b) Belowground Net Primary Productivity (BNPP), and (c) Net Primary Productivity (ANPP+BNPP)** of *Capsicum frutescens* under different fertilization treatments.

### 3.4. Quality traits increased with integrated nutrients

Results derived from ANOVA and Tukey's HSD test represented that our studied quality attributes (vitamin C content, capsaicin content, and SPAD reading) differed significantly ($p < 0.001$) among all individual fertilization treatments. The $T_4$ treatment recorded the highest vitamin C content (124.65±1.39 mg 100 g$^{-1}$), followed by $T_3$ (118.24±0.80), $T_0$ (112.97±1.11), $T_2$ (104.93±1.17) and $T_1$ (99.76±0.61) mg 100 g$^{-1}$ respectively (Fig 6a). Capsaicin content was also higher in $T_4$ (1.22±0.01%) compared to $T_3$ (1.13±0.01), $T_0$ (1.07±0.01), $T_2$ (1.00±0.01) and $T_1$ (0.91±0.01) % respectively (Fig 6b). Similarly, SPAD reading was highest under $T_4$ (48.61±0.14) treatment then $T_3$ (46.08±0.31), $T_2$ (43.06±0.38), $T_1$ (40.66±0.42) and $T_0$ (36.99±0.56) respectively (Fig 6c).

### 3.5. Significant associations between ANPP and vegetative reproductive traits

ANPP exhibited strong and statistically significant positive relations with both leaf number ($R^2 = 0.51$, $p = 0.02$) (Fig 7a) and flower number ($R^2 = 0.41$, $p = 0.014$) (Fig 7b), highlighting that greater vegetative and reproductive growth directly

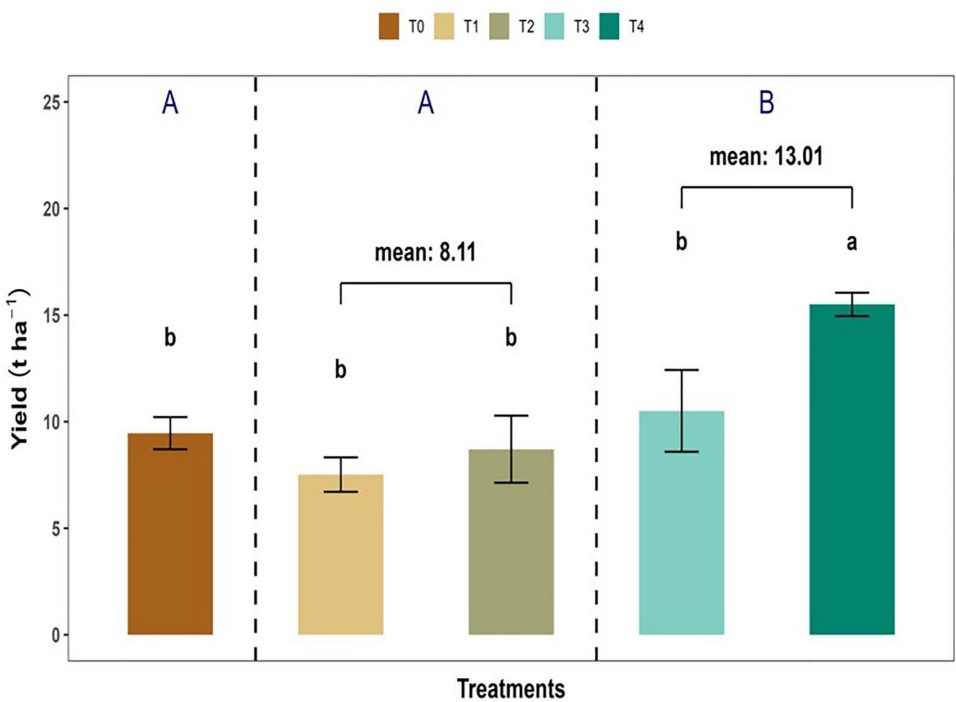

**Fig 5. Fruit yield (t ha⁻¹) of *Capsicum frutescens* under different fertilization treatments.**

enhanced aboveground biomass accumulation. Among the treatments, only $T_4$ (organic + ½ RDF) achieved the highest ANPP in relation to its reproductive traits (leaves: $R^2 = 0.57$, $p = 0.006$, and flowering: $R^2 = 0.68$, $p < 0.001$).

### 3.6. Yield is strongly influenced by vegetative and reproductive structures

A positive linear relationship was observed among yield, number of leaves ($R^2 = 0.34$, $p = 0.002$) (Fig 8a) and the number of flowering plants⁻¹ ($R^2 = 0.42$, $p = 0.037$) (Fig 8b), indicating their strong predictive value for yield performance. Among treatments, $T_2$ showed a significantly higher relation between leaf number versus yield ($R^2 = 0.80$, $p < 0.001$), as well as flowering versus yield ($R^2 = 0.82$, $p < 0.001$) respectively. $T_1$ also showed a moderate but significant relationship between yield and reproductive traits (leaves: $R^2 = 0.40$, $p = 0.041$, and flowering: $R^2 = 0.33$, $p = 0.046$). In contrast, the yield of $T_4$ did not show a significant correlation with plant reproductive traits (leaves: $R^2 = 0.12$, $p = 0.65$, and flowering: $R^2 = 0.28$, $p = 0.41$).

## 4. Discussion

### 4.1. Effect of mixed and mono fertilization on yield-contributing traits of chili

We aimed to determine whether mixed and mono fertilization enhances yield-contributing traits of chili, and we found that mixed fertilization (hereafter $T_3 + T_4$ combined) significantly increased most of the yield-contributing characters, such as plant height, leaf, flowering, and fruit number of chili. The vegetative traits, specifically plant height and leaf number, showed significant differences among different fertilization treatments, where the effect of mixed fertilization ($T_3 + T_4$) was consistently superior than control and organic fertilization treatment. These are important yield-contributing characteristics because they influence photosynthetic efficiency, carbon assimilation, and plant vigor [16,17]. In our experiment,

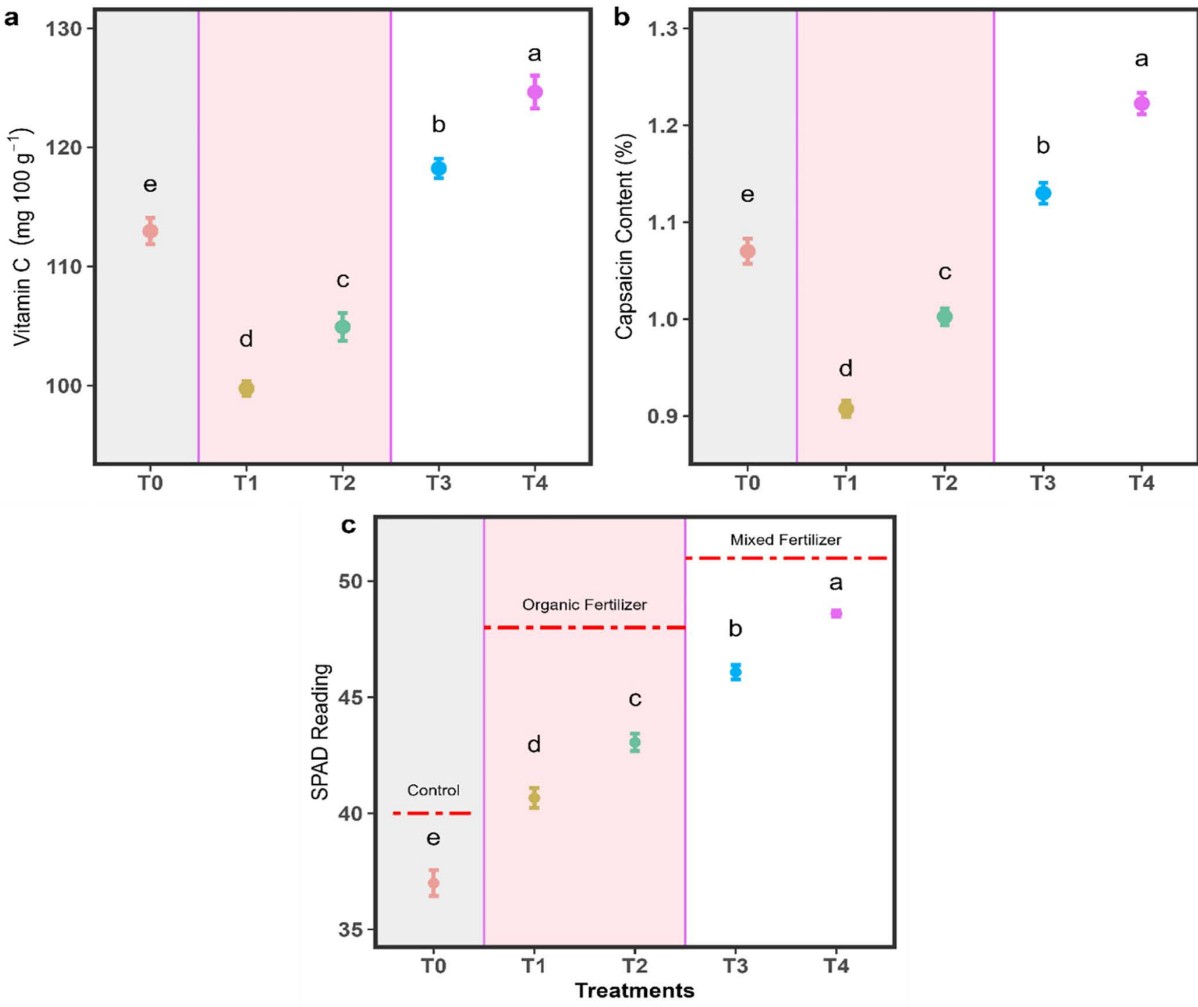

**Fig 6. Vitamin C, capsaicin content, and SPAD reading of *Capsicum frutescens* under different fertilization treatments.**

combined $T_3$ + $T_4$ treatment provides a dual advantage for plant height (Fig 3a) and leaf number (Fig 3b). Firstly, mixed fertilization results in immediate nutrient availability through inorganic fertilizers, and secondly, gradual nutrient release through organic amendments like *L. leucocephala* leaf litter. Previous studies have reported that the beneficial effects of integrated nutrient management (INM) promote plant growth and productivity. For instance, Malik et al. [46] and Zahid et al. [47] showed an improved plant height, branching, and canopy development in sweet pepper as well as cucumber when organic manure (e.g., poultry manure) was combined with reduced synthetic inputs. Our findings are also consistent with earlier studies indicating that synergistic application of organic and inorganic fertilizers enhances vegetative vigor due to improved nutrient availability, microbial activity, and soil structure [11, 48]. The reproductive traits, such as the number of flowers and fruits plant[-1] were also significantly increased through mixed fertilization (Fig 3). In our experiment, at 60 DAT, $T_4$ treatment exhibited nearly double the number of fruits set compared to the control and organic fertilization (Fig 3d). The increased flower formation and fruit set under integrated fertilization suggest a positive hormonal and nutritional state conducive to reproductive development. According to Sharma & Mittra [24], the presence of both macro and micro-nutrients in balanced proportions facilitates flowering and fruiting by supporting enzyme activity and hormonal synthesis.

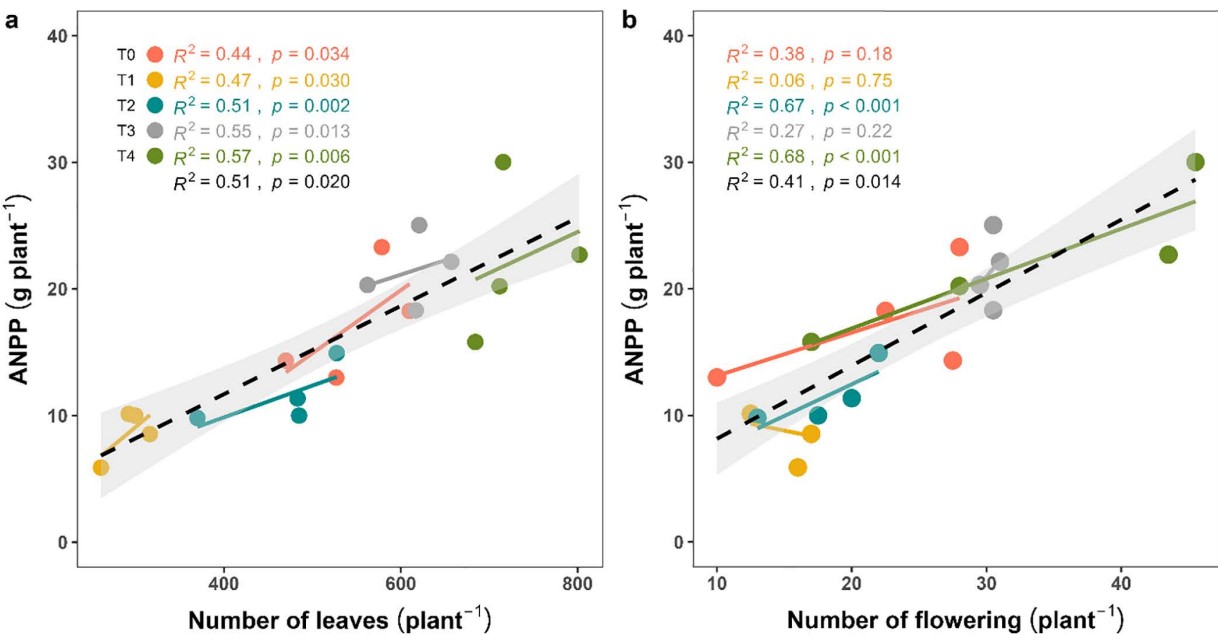

**Fig 7. Regression analysis between ANPP, vegetative (leaf number), and reproductive (flower number) traits of *Capsicum frutescens*.**

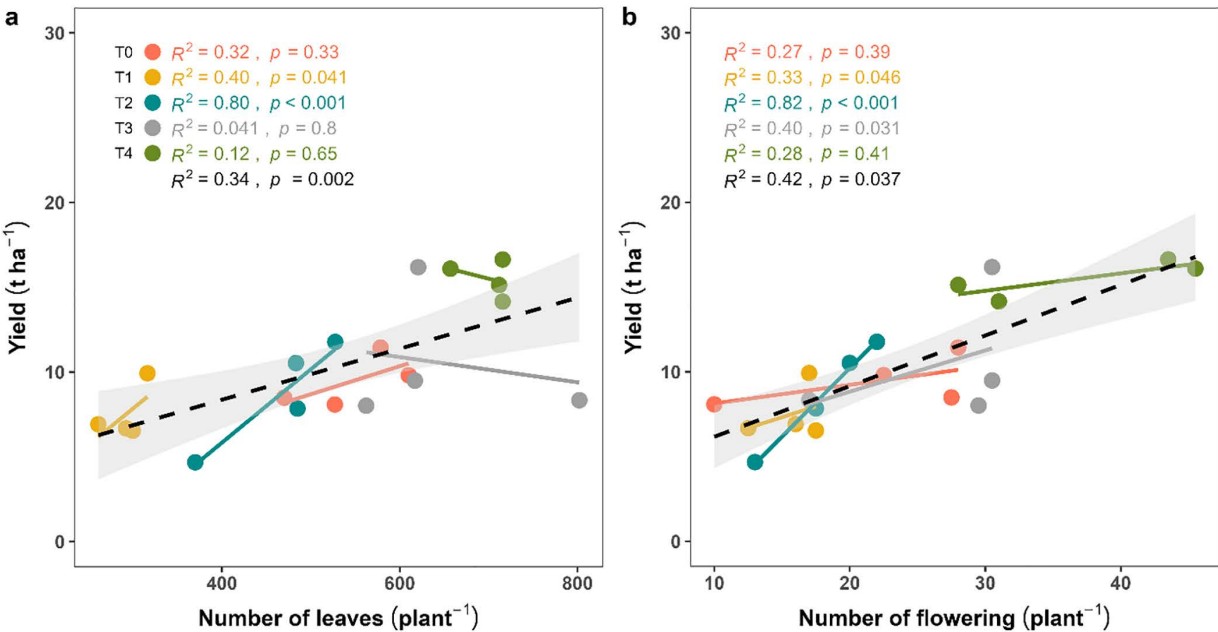

**Fig 8. Regression analysis showing the relationship between fruit yield and vegetative-reproductive traits (leaf and flower number) in *Capsicum frutescens*.**

Previous studies have also reported that the presence of organic matter, such as *L. leucocephala* leaf litter, may enhance rhizospheric microbial activity and nutrient-mediated hormonal processes that influence flower retention and fruit development [13,14]. However, direct measurements of microbial dynamics and hormone regulation would be required to confirm these pathways in future experiments. In our current experiment, such synergistic productivity outcomes will be difficult to achieve only through mono fertilization, especially when nutrient availability is either too rapid (in the case of synthetics) or too slow (in the case of organics). Hence, mixed fertilization offers a more buffered and physiologically aligned nutrient release pattern to support yield-contributing growth traits.

## 4.2. Effect of mixed and mono fertilization on NPP and fruit quality

Initially, we hypothesized that mixed and mono fertilization would increase the NPP and fruit quality of chili, and consistent with our hypothesis, we observed that the net primary productivity (NPP) was significantly greater under mixed fertilization ($T_3 + T_4$) compared to control ($T_0$) and organic fertilization ($T_1 + T_2$). Mixed fertilization ($T_3 + T_4$) treatment recorded the highest total NPP, indicating that integrated fertilization not only enhances aboveground shoot growth but also supports higher belowground root development (Fig 4c). These findings are in line with earlier studies by Jiang et al. [49] and Haque et al. [50], who observed that increased biomass accumulation under mixed fertilization (organic and inorganic) conditions due to improved nutrient use efficiency and soil nutrients dynamics. In our findings, increased BNPP under mixed fertilization may reflect enhanced root proliferation, root rhizosphere, microbial decomposition, and nutrient allocation, particularly carbon and nitrogen, which positively favored root biomass [51]. Nevertheless, these mechanisms were not directly quantified in the present study and should be examined in future work using targeted root and microbial measurements. Higher BNPP under mixed treatments suggests a healthy root system that can efficiently explore the soil matrix for water and nutrients. A similar scenario was observed in our experiment, indicating that mixed fertilization (e.g., *L. leucocephala* leaf litter & reduced synthetic fertilizer) responds more positively in case of BNPP (3.93 g plant$^{-1}$) compared to control (3.25 g plant$^{-1}$) and inorganic (1.95 g plant$^{-1}$) fertilizers (Fig 4b). Root development is strongly influenced by the physical and biological properties of the soil and have been reported to improve with the addition of organic matter [31,52]. Organic inputs are known to improve soil texture, moisture retention, and microbial colonization, which have been associated with upscaling root growth [53]. However, soil physical and microbial parameters were not directly measured here, and cannot be sufficiently verified within this experimental framework. Fruit yield was significantly enhanced under mixed fertilization, where $T_4$ achieved the highest yield (15.50 t ha$^{-1}$ and 63.9%) increase over the control (Fig 5). The higher yield under mixed fertilization may be associated with rapid nutrient availability from inorganic fertilizers together with nutrient release from organic inputs, which has been reported to improve nutrient use efficiency and reproductive development in previous studies [11,46]. However, nutrient release dynamics were not directly monitored in the present experiment and remain to be tested. Improved yield of chili under $T_4$ also aligns with the findings reported by Jiang et al. [49] and Gokul et al. [17], who reported that integrated nutrient management (INM) enhanced biomass and fruit productivity.

In support of these previous findings, we found that mixed fertilization strategy enhances productivity outcomes of chili under the present experimental conditions. Similar to productivity outcome, $T_4$ treatment showed outer performance in terms of fruit quality (e.g., vitamin C concentration, capsaicin content, and SPAD reading), and was driven primarily by mixed fertilization (Fig 6). Vitamin C and capsaicin are key quality indicators of chili, with implications for both nutrition and market value. Significant values of these biochemical compounds under $T_4$ suggested that integration of organic and inorganic fertilization supports optimal metabolic functioning (MF) of the plant. Previous literature indicates that balanced nutrient availability may enhance plants physiological and biochemical processes [15,17]. Considering limitation of our metabolic assessment, we suggested future research incorporating metabolic or enzymatic analyses, which would help clarify these responses under mixed fertilization strategy. The increased vitamin C under mixed fertilization is likely due to improved micronutrient availability, particularly potassium and magnesium, which play a critical role in ascorbate

biosynthesis [15]. Capsaicin synthesis is known to be elevated under balanced nitrogen regimes, which can be achieved through mixed fertilization [46]. Similarly, the enhanced SPAD reading indicates improved nitrogen assimilation and photosynthetic activity, further corroborating the physiological advantages of mixed input regimes [17]. Overall, the results confirm that mixed fertilization not only boosts NPP (Fig 4) but also enhances fruit quality (Fig 6), and offers a dual benefit for sustainable chili cultivation.

### 4.3. Relationships among morphological traits, biomass accumulation, and fruit yield

To explain our third objective, we performed a linear regression analysis, and the results revealed a significant positive relationship between vegetative and reproductive traits such as ANPP and fruit yield. Specifically, ANPP was positively related with leaf ($R^2 = 0.51$, $p = 0.020$) (Fig 7a) and flower number ($R^2 = 0.41$, $p = 0.014$) (Fig 7b), while both vegetative (leaf count) and reproductive (flower count) traits showed a similar positive trend with fruit yield (Fig 8). Monteith [54] proposed that crop yield is a function of photosynthetic efficiency and biomass partitioning, which are mediated by structural traits like leaf area and canopy architecture. Furthermore, Tilman et al. [55] and Agele et al. [56] have emphasized the role of nutrient balance in optimizing the source-sink relationship, thereby improving crop yield. Importantly, current study demonstrated that even partial replacement of synthetic fertilizers with organic sources, $T_3$ (leaf litter 93 g pot$^{-1}$ + ½ RDF) and $T_4$ (leaf litter 186 g pot$^{-1}$ + ½ RDF) can significantly improve biomass and yield without compromising physiological efficiency (Figs 4, 5, 7 and 8). This may have important implications for fertilizer management in resource-constrained settings where synthetic inputs are expensive or environmentally detrimental [57]. The strong positive relationships observed in this study suggest that mixed fertilization can serve as a strategy for improving nutrient use efficiency (NUE) and physiological productivity [58]. The integrative approach of nutrient supply provides a strength of fast-acting inorganic fertilizers in the soil, which directly promotes the quality of organic fertilizers and creates a robust, productive environment where plants can achieve their full physiological potential. This may contribute to higher yields probability but also promotes long-term sustainable and resilient chili production systems [59].

In our present study, the fertilizer treatments were not arranged as a complete factorial combination of organic and inorganic nutrient levels. Instead, the experiment has evaluated several specific treatment combinations that integrated *L. leucocephala* leaf litter with recommended mineral fertilizers. Because the design was not factorial, the independent main effects of organic and inorganic fertilization and their statistical interaction could not be estimated separately. Therefore, the observed improvements in plant growth, yield, and fruit quality should be interpreted as the overall response to the applied treatment combinations rather than isolated effects of individual nutrient sources. This limitation will open a potential source of future studies using a full factorial design, which allows a clearer separation of the individual and interactive contributions of organic and inorganic nutrient inputs.

## 5. Conclusions

The findings of this study demonstrate that mixed fertilization strategies have a more substantial positive effect on growth, productivity, and quality of chili (*Capsicum frutescens*) than mono-fertilization. Three key outcomes mainly emerged from this study. **Firstly,** vegetative traits such as plant height and leaf number were consistently enhanced under mixed fertilization, suggesting improved nutrient availability and plant health. **Secondly,** aboveground and belowground biomass accumulation (representing the ANPP and BNPP) were notably higher in integrated nutrient treatments, indicating optimized NPP. **Thirdly,** essential fruit quality attributes, including vitamin content, capsaicin concentration, and SPAD reading, were significantly improved under mixed fertilization, highlighting its positive impact on metabolic and biochemical functions. **Additionally,** significant positive relationships were observed between yield-contributing traits with both ANPP and fruit yield, confirming that vegetative and reproductive traits strongly determine the overall productivity. Overall, our study highlighted that mono fertilization, whether organic or inorganic limit the productivity and quality benefits when compared to mixed fertilization. Hence, integrated nutrient management (considering both organic and inorganic fertilizers)

should be conceived as a climate-resilient and resource-efficient practice for chili cultivation in Bangladesh and other geographic areas with a similar agroecological environment. Despite our current intensive research, critical treatment limitations still exist (absence of a complete factorial combination of organic and inorganic nutrient treatment), which limits our understanding. To overcome these bottlenecks, we call for more factorial experimental designs in the future that explicitly test these issues.

## Supporting information

**S1 File. R scripts.**
(DOCX)

**S2 File. Manuscript data.**
(XLSX)

## Acknowledgments

The authors would like to thank the senior field laboratory technician, Mr. Aminul Islam, for his assistance with soil preparation, sowing, and intercultural operations during this research. The authors also gratefully acknowledge the reviewers for their valuable comments and suggestions, which helped improve this manuscript.

## Author contributions

**Conceptualization:** Hossain Md Dalim, Md Golam Jilani Helal.

**Data curation:** Hossain Md Dalim, Minhazul Kashem Chowdhury, Md. Shahariar Jaman.

**Formal analysis:** Hossain Md Dalim, Md. Shahariar Jaman.

**Funding acquisition:** Md Golam Jilani Helal, Md. Shahariar Jaman.

**Investigation:** Md Golam Jilani Helal, Md. Shahariar Jaman.

**Methodology:** Hossain Md Dalim, Md Golam Jilani Helal, Md. Shahariar Jaman.

**Project administration:** Md Golam Jilani Helal, Md. Shahariar Jaman.

**Resources:** Shoaib Rahman, Md. Mohi Uddin Sujan Chowdury.

**Software:** Hossain Md Dalim, Md Golam Jilani Helal, Md. Shahariar Jaman.

**Supervision:** Md. Shahariar Jaman.

**Validation:** Minhazul Kashem Chowdhury, Shoaib Rahman, Sohel Rana Mazumder, Md. Ismail Hossain, Md. Mohi Uddin Sujan Chowdury.

**Visualization:** Sohel Rana Mazumder, Md. Ismail Hossain.

**Writing – original draft:** Hossain Md Dalim.

**Writing – review & editing:** Hossain Md Dalim, Md Golam Jilani Helal, Minhazul Kashem Chowdhury, Md. Shahariar Jaman.

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
