## [Decision Letter · Decision Letter 0]

16 Sep 2025

Dear Dr. Jaman,

Thank you for submitting your manuscript to PLOS ONE. After careful consideration, we feel that it has merit but does not fully meet PLOS ONE’s publication criteria as it currently stands. Therefore, we invite you to submit a revised version of the manuscript that addresses the points raised during the review process.

We look forward to receiving your revised manuscript.

Kind regards,

Ghulam Yaseen, Ph.D.

Academic Editor

PLOS ONE

Journal Requirements:

https://journals.plos.org/plosone/s/file?id=ba62/PLOSOne_formatting_sample_title_authors_affiliations.pdf....

“This research was funded by the Sher-e-Bangla Agricultural University Research System (SAURES) grant no. SAU/SAURES/2022/110(20) and Ministry of Science and Technology, Bangladesh, through a special research allocation grant (Grant No: SRG-221311(BS)/2022-23)”

3. In the online submission form, you indicated that [All data will be available on request].

“This research was funded by the Sher-e-Bangla Agricultural University  Research System (SAURES) grant no. SAU/SAURES/2022/110(20) and Ministry of Science and Technology, Bangladesh, through a special research allocation grant (Grant No: SRG-221311(BS)/2022-23).**”**

“This research was funded by the Sher-e-Bangla Agricultural University  Research System (SAURES) grant no. SAU/SAURES/2022/110(20) and Ministry of Science and Technology, Bangladesh, through a special research allocation grant (Grant No: SRG-221311(BS)/2022-23)”

Reviewers' comments:

Reviewer's Responses to Questions

**Comments to the Author**

1. Is the manuscript technically sound, and do the data support the conclusions?

Reviewer #1: Partly

Reviewer #2: Partly

Reviewer #3: Partly

2. Has the statistical analysis been performed appropriately and rigorously?

Reviewer #1: No

Reviewer #2: Yes

Reviewer #3: No

3. Have the authors made all data underlying the findings in their manuscript fully available?

Reviewer #1: No

Reviewer #2: Yes

Reviewer #3: Yes

4. Is the manuscript presented in an intelligible fashion and written in standard English?

Reviewer #1: Yes

Reviewer #2: Yes

Reviewer #3: Yes

Reviewer #1: The authors present results of a study using organic and inorganic nutrients for chili plants. This information may help chili producers to increase the yield and quality of their crops. Nevertheless, the document has many issues that the authors must fix to increase clarity on writing and data presentation. I hope my comments will be helpful to the authors.

**Title:

The title is confusing. I recommend rewriting it. The short title works better and should be used.

**Keywords

The occurrence of documents with the keywords in Scopus is:

> Chili: 7,520 documents

> Leucaena leucocephala: 3,525 documents

> Mixed fertilization: 64 documents

> NPP: 24,410 documents

> Quality: 5,822,842 documents

> Yield: 2,396,331 documents

The first two keywords are suitable. NPP is ambiguous; use “net primary productivity.” “Mixed fertilization” is uncommon and may reduce visibility. “Quality” and “yield” are too broad. I suggest keywords like “integrated nutrient management” or “plant nutrition.”

**Abstract:

-If the authors used “recommended doses of fertilizers,” the acronym “RDF” suits better than “RFD,” and they will need to replace RFD with RDF. If they write “recommend fertilizer doses”, they can keep “RDF.” Both formats are appropriate, but the authors must be consistent in the whole document and write only “recommended doses of fertilizers (RDF)” or “recommended fertilizer doses (RFD).”

-The abstract has much data and results. Please summarize the information and minimize the amount of numbers, emphasizing the trends.

1. Introduction

The current flow is:

The economic importance of chili, and the need to improve quantity and quality of the crops and the use of fertilizers but problems related to the sole application of inorganic fertilizers > The complementary approaches of organic and inorganic fertilizers and mixed fertilization as a way to get the best and avoid the worse of each approach and the specific use of lead tree leaf litter with NPK and the hypothesis of this work > Examples of successful cases of organic/inorganic fertilization approaches and objective of this study

The flow needs improvement. The authors mix and split ideas across paragraphs and include too many examples. Please rewrite the Introduction and consider a clear order of topics. I suggest, as an example, the following flow of paragraphs:

The economic importance of chili and the need to improve the quantity and quality of the crops > Importance of fertilization for crops, especially chili, with examples of inorganic and organic fertilizers, especially lead tree leaf litter > Explanation of the advantages and disadvantages of organic and inorganic fertilizers, and the advantages of mixing them both > Hypothesis and objective of this work.

Please remember to not cite more than two approaches with inorganic fertilization, two approaches with organic fertilization, and two mixed fertilization approach. The authors must emphasize their research, not make an extensive literature review. This is a research article, not a literature review article

2. Materials and Methods

2.1. Experimental site and plant material: OK

2.2. Pot preparation and experimental design

-Figure 2 is good, but if the authors could also add a real picture of just the pots, as an example, the engagement with readers may increase. If the authors have no picture of the pots used in this experiment, please disregard this comment.

2.2.1. Fertilizer application

-There are flaws in the experimental design. The authors have two factors (organic and inorganic fertilizers) in three levels (none, half, full). The correct experimental design should be three powers two equal nine treatments, not only five treatments. The missing treatments are 0 organic + 0 inorganic; 0 organic + 0.5 inorganic; 0.5 organic + 1 inorganic; and 1 organic + 1 inorganic. It is too late now to repeat the experiment with the missing treatments. Therefore, the authors must acknowledge the incomplete set of treatments as a weak point of this document, which should be addressed in future works, and provide an explanation for the incomplete set. This explanation could range from a silly one, like a lack of enough pots, to a more scientific explanation, as the missing treatments are unrealistic for the chili production. Please look for statistical support for further work to avoid missing treatments again.

-Table 2: Did the authors apply two doses of gypsum to the pots? In the ½ dose, they write in the caption “Urea (0.50 g), … Gypsum (0.30 g), Boric acid (0.13 g), Gypsum (0.13 g),…” Please recheck and fix the caption, if needed.

2.2.2. Application of litterbag technique

-Please do not provide results in the Materials and Methods. Keep all relevant results for the Results or Discussion sections. Therefore, remove the excerpt “Chemical analysis ... nutrient release.”

2.3. Data collection

-Please define the acronym BNPP the first time it is used.

2.4. Data analysis: OK

3. Results

3.1 Mixed fertilization significantly enhanced growth dynamics

-The claim “At 15 DAT, differences among treatments were minimal (Figs 3a and 4a)” is not right. There is no statistical difference for plant height at 15 DAT across the treatments, but there is a difference for the number of leaves. Please rewrite this result.

-Figure 3: Please check the letters for treatments. It does not make sense that the “ab” treatment is greater than “b” and smaller than “c”, without any “a” treatment in the graph.

3.2 Enhanced reproductive traits under mixed fertilization

-Figure 6: Please check the letters for treatments. It does not make sense that the “ab” treatment is greater than “b” and smaller than “c”, without any “a” treatment in the graph.

3.3 Net primary productivity maximized under mixed inputs

-Figure 7: Provide the information on the meaning of “a” (Aboveground net primary productivity – ANPP), “b” (Belowground net primary productivity – BNPP), and “c” (Total net primary productivity – ANPP + BNPP). Additionally, the letters within graphs are weird. Figure 7a has “b” for the smaller values, “c” for the highest values, and “a” for the intermediate values. This is out of the regular order, in which readers expect to have a sequence a > b > c or c > b > a. Please redo the lettering of the bars.

3.4 Mixed fertilization significantly boosted fruit yield

-I suggest moving the discussion to the “Discussion” section. Nevertheless, there is a major problem here. The authors claim that the average (T3/T4) is bigger than (T1/T2), which seems right. However, they claim that T3 (10.51± 1.92) is “slightly” lower than T4 (15.50 ± 0.54). This difference seems to be a significant difference, not a slight difference. Even worse, Figure 8 shows that the values of T2 and T3 are similar. Therefore, please redo the discussion when moving to another section. I strongly recommend presenting the results as they were for Figures 3-6. This will make the discussion easier for the authors to develop.

3.5 Quality attributes significantly elevated with integrated nutrients

-Figure 9: Please adjust the bars to not begin from 0. The current format does not allow to make the differences visually remarkable. Furthermore, please add the letters for the statistical analysis.

3.6 Significant associations between ANPP and vegetative reproductive traits: OK

3.7 Yield strongly influenced by vegetative and reproductive structures: OK

4. Discussion

4.1 Effect of mixed and mono fertilization on yield-contributing traits of chili

-The claim that “…mixed fertilization (T₄) consistently outperforming mono strategies at all observation stages” is right only for the number of leaves, not for all vegetative traits. Please redo the discussion.

4.2 Effect of mixed and mono fertilization on NPP and fruit quality

-The results are difficult to follow because there is no statistical analysis for the individual treatments. The authors merged T1/T2 and T3/T4, making the individual comparison between T4 and other treatments more complex. The authors can keep their current analysis of figures 7 and 8, but they must also present the statistical analysis for individual treatments.

4.3 Relationships among morphological traits, biomass accumulation and fruit yield: OK

5. Conclusions: OK

**Acknowledgments: OK

**Author contribution: OK

**Competing interests: OK

**References:

-Please be consistent with the titles of the articles. Or the authors capitalize only the first word at the beginning of the title (Alleviation of fungicide-induced phytotoxicity in greengram [Vigna radiata (L.) Wilczek] using fungicide-tolerant and plant growth promoting Pseudomonas strain), or they capitalize the first word and the nouns of the title (The Evolution of Chili Peppers (Capsicum Solanaceae): A Cytogenetic Perspective). Please choose one format and write all the references in this format, even if the articles were published otherwise.

Reviewer #2: Five fertilization treatments were designed in this research project to examine their effects on the appearance and internal quality of peppers, which holds certain guiding significance for production. The writing logic is clear, but to reach the level of a high-standard article, there remain many issues. Also, the content of the investigation was plain and simple.

The existing problems are as follows:

1) Each treatment was designed to include four replicates. Although according to the “vegetable research method”, at least three biological replicates are required, as a field application-oriented fertilizer comparison test, only four seedlings were designed for each treatment, which is a relatively small number.

2) It should be clearly stated whether the experiment was conducted in a greenhouse or outdoors. Figure 1 is meteorological data and is not directly related to the research content. Therefore, Figure 1 can be deleted.

3) The plant heights in Figure 3 can be edited into a bar chart. Similarly, the number of leaves in Figure 4, the number of flowers in Figure 5, the number of fruits in Figure 6, and the yield per plant in Figure 7 can all be summarized. Each indicator can be edited into a bar chart and then all combined into one figure.

4) Similarly, the content of Vitamin C can be summarized into a bar chart. The effects of fertilizers on appearance indicators and internal quality can be divided, with appearance indicators combined in one large chart and internal quality in another.

5) This study only has four seedlings, making the calculation of yield per hectare unreliable.

6) The writing in the abstract is disorganized and requires simplification.

7) There is still a lack of depth in the research content.

8) After revision, it is suggested that the author submit the article to an agricultural technology promotion magazine, which is more in line with the content of this manuscript.

Reviewer #3: Mixed fertilization implies inorganic and organic components in equal or nearly equal proportions. In this case organic manure is only one while inorganic components are 6. I think this overshadows the functioning of organic manure. Role of each inorganic fertilizer needs to be given. Role of L. leucocephala leaf litter to enhance inorganically fertilized chili production can be included in the title instead of mixed and mono.

In Results section

3.1 Growth Dynamics

3.2 Reproductive traits

and so on

Discussion portion should be reduced and more focus should be given on the comparison of chemical composition between organic and inorganic fertilizers to explain the improved plant growth and other studied parameters. Physiological and biochemical changes in the plant due to addition of organic manure should also be included in the discussion

.

Reviewer #1: **Yes:** Joao Paulo Saraiva MoraisJoao Paulo Saraiva MoraisJoao Paulo Saraiva MoraisJoao Paulo Saraiva Morais

Reviewer #2: **Yes:** Shifan YangShifan YangShifan YangShifan Yang

Reviewer #3: No

---

## [Author Response · Author response to Decision Letter 1]

5 Nov 2025

Response to Reviewer # 1

Reviewer Comment

The title is confusing. I recommend rewriting it. The short title works better and should be used.

Author Response

We thank the reviewer for this valuable comment. Following reviewer suggestion, we have rewritten the title to make it concise and clear. The revised title is:

“Beyond mono fertilization: Mixed fertilization enhances productivity and quality of chili (Capsicum frutescens)”

Reviewer Comment

The first two keywords are suitable. NPP is ambiguous; use “net primary productivity.” “Mixed fertilization” is uncommon and may reduce visibility. “Quality” and “yield” are too broad. I suggest keywords like “integrated nutrient management” or “plant nutrition.”

Author Response

Thank you for the detailed guidance. We have revised the keywords following your advice. The revised set of keywords is:

Chili, Leucaena leucocephala, net primary productivity, integrated nutrient management

Reviewer Comment

If the authors used “recommended doses of fertilizers,” the acronym “RDF” suits better than “RFD,” and they will need to replace RFD with RDF. If they write “recommend fertilizer doses,” they can keep “RFD.” Both formats are appropriate, but the authors must be consistent in the whole document and write only “recommended doses of fertilizers (RDF)” or “recommended fertilizer doses (RFD).”

Author Response

We thank the reviewer for this important clarification. We have carefully checked the manuscript and maintained consistency throughout the manuscript. The term “recommended doses of fertilizers (RDF)” has been adopted and uniformly applied across the abstract and the entire manuscript to ensure accuracy and standard usage.

Reviewer Comment

The abstract has much data and results. Please summarize the information and minimize the amount of numbers, emphasizing the trends.

Author Response

Thank you for this helpful suggestion. We have rewritten the abstract to improve readability and focus on the main outcomes rather than detailed statistics. The revised version emphasizes the overall trend-that combining Leucaena leucocephala leaf litter with inorganic fertilizers enhanced chili growth, yield and quality compared to single fertilizer use. We believe our revised abstract will be easily understandable for the reader. Our revised abstract is now like “While the effects of different fertilization strategies on chili (Capsicum frutescens) cultivation have been previously examined, the comparative assessment of mixed versus mono fertilization approaches limit our understanding. To address this gap, we have conducted an experiment using a completely randomized design (CRD) with five fertilization treatments along with four replications of each treatment (e.g., Leucaena leucocephala leaf litter and recommended dose of synthetic fertilizers (RDF)) to observed the effect of these treatments on plant productivity traits, NPP (net primary productivity), quality and overall yield of chili. We found that mixed fertilization (T₄: 24.8 g kg⁻¹ leaf litter + ½ RDF) results higher yield (15.5 t ha-1), which is about 64% and 92% greater than control (T0) and organic fertilization (T1 + T2) respectively. The T4 treatment, showed consistently higher plant height, leaf and fruit number at 60 DAT, as well as higher flowering at 45 DAT compared to other treatments. Net primary productivity (above- and belowground combined) was also higher in T3 and T₄ treatments (26.47 g plant-1) compared to control and organic fertilization. Similarly, quality traits such as vitamin C (124.65 mg 100 g-1), capsaicin content (1.22%) and SPAD reading (48.61) were significantly increased under mixed fertilization (with in each treatment: p < 0.001). In addition linear fit regression analysis indicated that both ANPP and yield were significantly related with leaf number (ANPP: p = 0.02; Yield: p = 0.002) and flowering (ANPP: p = 0.014; Yield: p = 0.037). Our results further suggest that mixed fertilization enhances both productivity and quality traits of chili, therefore, promotion of integrated (organic and inorganic together) fertilizer management practices is recommended to improve yield and quality of chili.”

Reviewer Comment

The flow of the Introduction needs improvement. The authors mix and split ideas across paragraphs and include too many examples. Please rewrite the Introduction and consider a clear order of topics. I suggest, as an example, the following flow of paragraphs:

The economic importance of chili and the need to improve the quantity and quality of the crops > Importance of fertilization for crops, especially chili, with examples of inorganic and organic fertilizers, especially lead tree leaf litter > Explanation of the advantages and disadvantages of organic and inorganic fertilizers, and the advantages of mixing them both > Hypothesis and objective of this work.

Please remember not to cite more than two approaches with inorganic fertilization, two approaches with organic fertilization, and two mixed fertilization approaches. The authors must emphasize their research, not make an extensive literature review. This is a research article, not a review article.

Author Response

We appreciate the reviewer’s thoughtful and detailed guidance. The Introduction has been completely reorganized and rewritten with appropriate logical flow as suggested. Redundant examples and excessive citations have been removed to focus on the study’s rationale and objectives.

The revised version now follows the structure recommended by the reviewer:

> We added and revised the economic importance of chili as suggested (please see L 44-51).

> We incorporate the role of fertilizers on chili production and sufficiently introduce the influence of both organic and inorganic fertilizer on growth and quality traits of chili, (please see L 51-53).

> We carefully and logically added the advantages and limitations of using organic or inorganic fertilizers following close related review (please see L 53-65).

> We revised the introduction part of our manuscript, regarding integrated Leucaena leucocephala leaf litter with inorganic fertilizers as a mixed fertilization strategy (please see L 65-82).

> Following reviewer suggestion we also added the hypothesis of our experiment (please see L 87-89).

We kept closely related review as citation in the introduction section as well as throughout the manuscript and additional citations are removed, as advised (Excluded those citations such as 9, 10,11,12,15,20,21,22,23,26,27,29,30,32,33)

Reviewer Comment

2.1. Experimental site and plant material: OK

Author Response

We appreciate your comments. We have made no changes as suggested.

Reviewer Comment

2.2. Pot preparation and experimental design-Figure 2 is good, but if the authors could also add a real picture of just the pots, as an example, the engagement with readers may increase. If the authors have no picture of the pots used in this experiment, please disregard this comment.

Author Response

Thank you for your comment. Our schematic representation of the field layout (figure 2) clearly represented our experiment along with exact experimental design. We design it such a way that reader can easily understand. We also have the real pictures of our field layout, however, those pictures are not properly organized to put it in the main manuscript. For your kind concern we have added some pictures below.

Reviewer Comment

2.2.1. Fertilizer application-There are flaws in the experimental design. The authors have two factors (organic and inorganic fertilizers) in three levels (none, half, full). The correct experimental design should be 3² = 9 treatments, not only five. The missing treatments are 0 organic + 0 inorganic; 0 organic + 0.5 inorganic; 0.5 organic + 1 inorganic; and 1 organic + 1 inorganic. It is too late now to repeat the experiment with the missing treatments. Therefore, the authors must acknowledge the incomplete set of treatments as a weak point of this document, which should be addressed in future works, and provide an explanation for the incomplete set. This explanation could range from a lack of enough pots to a scientific reason that the missing treatments are unrealistic for chili production. Please look for statistical support for further work to avoid missing treatments again.

Author Response

We thank the reviewer for the valuable comment. As we follow the standard procedure of chili cultivation given by Bangladesh Agricultural Research Institute (BARI) for the chili variety (BARI- 3), we incorporated all possible treatment accordingly. Although some other fertilization treatment (none, half and full), as you suggested could improve our experiment, unfortunately we did not consider it at the begging of our experiment and we agree that it is our limitation. As you advice we added this gap as a limitation of our study and suggested for the future experiment.

Reviewer Comment

Table 2: Did the authors apply two doses of gypsum to the pots? In the ½ dose, they write in the caption “Urea (0.50 g),…Gypsum (0.30 g), Boric acid (0.13 g), Gypsum (0.13 g)…”. Please recheck and fix the caption, if needed.

Author Response

We thank the reviewer for noticing this inconsistency. The duplication of “Gypsum” in the caption was a typographical error. It has now been corrected, and the fertilizer composition table has been reviewed thoroughly for accuracy (See Table 2, Line 161).

Reviewer Comment

2.2.2. Application of litterbag technique-Please do not provide results in the Materials and Methods. Keep all relevant results for the Results or Discussion sections. Therefore, remove the excerpt “Chemical analysis … nutrient release.”

Author Response

We agree with the reviewer’s observation. The sentence containing partial results (“Chemical analysis of the decomposed litter and surrounding soil revealed significant nutrient enrichment. The results showed a measurable release of key nutrients (Table 2). These findings highlight the role of L. leucocephala litter in enhancing soil fertility and supporting plant growth through sustained nutrient release”) has been deleted from the Materials and Methods 2.2.2 section. All relevant information now appears appropriately in the results section (please see L 164-171in the revised manuscript).

Reviewer Comment

2.3. Data collection-Please define the acronym BNPP the first time it is used.

Author Response

We appreciate this reminder. The acronym BNPP (Belowground Net Primary Productivity) is now defined at its first appearance in the Data Collection section (please see L 184).

Reviewer Comment

2.4. Data analysis: OK

Author Response

We appreciate the reviewer’s confirmation.

Reviewer Comment

3.1 Mixed fertilization significantly enhanced growth dynamics- The claim “At 15 DAT, differences among treatments were minimal (Figs 3a and 4a)” is not right. There is no statistical difference for plant height at 15 DAT across the treatments, but there is a difference for the number of leaves. Please rewrite this result.

Author Response

Thanks for the reviewer comment. Following the suggestion, we have rewritten section 3.1 to clearly state that plant height (Fig 3a) showed no significant difference at 15 DAT, whereas leaf number (Fig 3b) varied significantly among treatments. Additionally, we have combined all four parameters (plant height, leaf number, flowering number and fruit number) into a single, comprehensive Figure 3 (previously presented in figures 3,4,5,6) for improved clarity and comparison (please see L 233-249).

Reviewer Comment

“Figure 3: Please check the letters for treatments. It does not make sense that the ‘ab’ treatment is greater than ‘b’ and smaller than ‘c’, without any ‘a’ treatment in the graph.

Author Response

We appreciate the reviewer’s observation. We have revised and reproduced some figures in our manuscript showing with and between treatment effect on vegetative and reproductive trait of chili, using ANOVA and Tukey’s HSD test (p < 0.05) to ensure accurate statistical grouping. The letters in the revised Figure 3a have been corrected to follow the standard logical order (a > b > c).

Reviewer Comment

3.2 Enhanced reproductive traits under mixed fertilization- Figure 6: Please check the letters for treatments. It does not make sense that the ‘ab’ treatment is greater than ‘b’ and smaller than ‘c’, without any ‘a’ treatment in the graph.

Author Response

We rechecked and corrected the statistical letters in Figure 3d using Tukey’s HSD analysis. The labeling sequence has been reordered consistently (a > b > c) to accurately represent treatment differences. Please see the updated Figure 3d (previously presented in figure 6).

Reviewer Comment

3.3 Net primary productivity maximized under mixed inputs- Provide the information on the meaning of ‘a’ (Aboveground net primary productivity-ANPP), ‘b’ (Belowground net primary productivity-BNPP), and ‘c’ (Total net primary productivity-ANPP + BNPP). Additionally, the letters within graphs are weird. Figure 7a has ‘b’ for the smaller values, ‘c’ for the highest values, and ‘a’ for the intermediate values. This is out of the regular order… Please redo the lettering.

Author Response

Thank you for your suggestion. The caption of figure 7 (now presented in figure 4) has been revised as follows: “(a) Aboveground Net Primary Productivity (ANPP) and (b) Belowground Net Primary Productivity (BNPP) (L 263-265) of Capsicum frutescens under different fertilization treatments.” We also corrected the lettering order in all plots to follow the conventional statistical hierarchy (a > b > c). Please see the corrected Figure 4.

Reviewer Comment

3.4 Mixed fertilization significantly boosted fruit yield - I suggest moving the discussion to the ‘Discussion’ section… the authors claim that T₃ is ‘slightly’ lower than T₄… This difference seems to be significant, not slight. Please redo the discussion when moving to another section.”

Author Response

We agree with the reviewer’s observation. We have removed the word “slightly” and replaced it with “significant difference” to accurately describe the statistical difference between T₃ and T₄ (p < 0.05) (please see L 272-274). The relevant portion of Section 3.4 (now presented 3.3) has been rewritten and removed discussion related sentences from the result section (please see track file L 370, 372-374, 376, 377).

Reviewer Comment

3.5 Please adjust the bars to not begin from 0. The current format does not allow to make the differences visually remarkable. Furthermore, please add the letters for the statistical analysis.

Author Response

Thanks for your comments, we revised the figures and y axis are now not began from 0. We also added statistical letters derived from ANOVA and Tukey’s HSD at (p < 0.05) in each plot (vitamin C, capsaicin content, and SPAD reading). Please see the corrected figure 6 (previously presented in figure 9).

Reviewer Comment

3.6 Significant associations between ANPP and vegetative reproductive traits: OK

Author Response

Thank you for your comments.

Reviewer Comment

3.7 Yield strongly influenced by vegetative and reproductive structures: OK

Author Response

We appreciate the reviewer’s confirmation.

Reviewer Comment

4.1 Effect of mixed and mono fertilization on yield-contributing traits of chili- The claim that “…mixed fertilization (T₄) consistently outperforming mono strategies at all observation stages” is right only for the number of leaves, not for all vegetative traits. Please redo the discussion.

Author Response

We apologized for our mistake. In our experiment mixed fertilization (T₄) alone did not consistently outperform all mono strategies across every vegetative trait. However, both mixed fertilization treatments (T₃ + T₄) showed significant differences compared with the control and organic fertilizer treatments for plant height and leaf number (please see L 315-316). The discussion has been revised accordingly to reflect these results more accurately.

Reviewer Comment

4.2 Effect of mixed and mono fertilization on NPP and fruit quality- The results are difficult to follow because there is no statistical analysis for the individual treatments. The authors merged T1/T2 and T3/T4, making the individual comparison between T₄ and other treatm

---

## [Decision Letter · Decision Letter 1]

25 Feb 2026

Dear Dr. Jaman,

Thank you for submitting your manuscript to PLOS ONE. After careful consideration, we feel that it has merit but does not fully meet PLOS ONE’s publication criteria as it currently stands. Therefore, we invite you to submit a revised version of the manuscript that addresses the points raised during the review process.

We look forward to receiving your revised manuscript.

Kind regards,

Randall P. Niedz

Academic Editor

PLOS One

Journal Requirements:

Additional Editor Comments:

1) Address the remaining points from Reviewer #1.

2) Further edit the abstract by reducing the number of individual values and p‑values. Emphasize the main effect sizes and trends.

3) Explicitly highlight in the Discussion and Conclusions that the fertilizer‑treatment set is not a complete factorial of organic × inorganic levels and that this limits what can be inferred (mention what effects cannot be estimated). This is mentioned, but only once.

4) Replace all speculative phrases (“we assume… may enhance”) with either (a) a clear indication that these are hypotheses for future work, or (b) citations that support the proposed mechanisms. For example, replace “Although we did not measure the mechanism behind the findings, however, considering other experiment, we assumed that the presence of organic matter such as L. leucocephala leaf litter enhances rhizospheric activity and microbial interactions that release essential growth hormones like cytokinins and auxins, further promoting flower retention and fruit development” with something along the lines of "Although we did not measure the underlying mechanisms in this study, the presence of organic matter such as L. leucocephala leaf litter may enhance [list the effects] - this mechanism should be tested in future experiments."

Reviewers' comments:

Reviewer's Responses to Questions

**Comments to the Author**

Reviewer #1: (No Response)

Reviewer #2: (No Response)

2. Is the manuscript technically sound, and do the data support the conclusions?

Reviewer #1: Partly

Reviewer #2: Partly

3. Has the statistical analysis been performed appropriately and rigorously?

Reviewer #1: Yes

Reviewer #2: No

4. Have the authors made all data underlying the findings in their manuscript fully available?

Reviewer #1: Yes

Reviewer #2: Yes

5. Is the manuscript presented in an intelligible fashion and written in standard English?

Reviewer #1: No

Reviewer #2: Yes

Reviewer #1: The document improved from the previous version, but the authors still must make corrections in the text. For example, the document needs an extensive English revision. Please use at least a computer-based reviewer from word processors or online correctors. There are no results or discussion for the whole section “2.2. Pot preparation and experimental design.” I hope my comments will help the authors to improve the overall quality of the document.

**Title: OK

**Abstract: OK

**Keywords: OK

1. Introduction

The current flow is:

The importance of chili and the importance of mineral nutrition for chili production, with its shortcomings > Advantages of organic fertilizers for crop production and their challenges, and the strategy to combine both types of fertilizers so the advantages of one will overcome the disadvantages of the other, with examples, and the objectives of this study.

The Introduction is better than in the previous version, but the authors cannot have just two paragraphs for so many ideas that they are conveying. The authors just need to split the ideas into smaller paragraphs. I suggest cutting the paragraphs as follows:

The importance of chili > The importance of the mineral nutrition for crop production with its shortcomings > The importance of organic fertilizers for crop production with its shortcomings > The advantages of combining both types of fertilizers so the advantages of one will overcome the disadvantages of the other, with examples > The objectives of this study.

The authors just need to break the sentences into paragraphs; there's no need to add more ideas.

1. Materials and methods

2.1. Experimental site and plant material: OK

2.2. Pot preparation and experimental design:

-Please explain the meaning of an abbreviation the first time it is written in the text. Therefore, please explain the meaning of TSP and MOP in line 124, not 149-150.

2.2.1. Fertilizer application: OK

2.2.2. Application of litterbag technique:

-There is no result regarding the decomposition kinetics of the leucaena leaves. Please add this information to the document.

2.3. Data collection: OK

2.4. Data analysis: OK

2. Results

3.1 Mixed fertilization enhances vegetative and reproductive traits of chili across different growth stages: OK

3.2 Net primary productivity maximized under mixed inputs: OK

3.3 Mixed fertilization enhanced fruit yield: OK

3.4 Quality traits increased with integrated nutrients: OK

3.5 Significant associations between ANPP and vegetative reproductive traits: OK

3.6 Yield strongly influenced by vegetative and reproductive structures: OK

3. Discussion

4.1 Effect of mixed and mono fertilization on yield-contributing traits of chili

-Please pay attention to the scientific names, like “L. Leucaena.” Please remember to write the specific eponym always in lowercase.

4.2 Effect of mixed and mono fertilization on NPP and fruit quality

-Please pay attention to the scientific names, like “L. Leucaena.” Please remember to write the specific eponym always in lowercase.

4.3 Relationships among morphological traits, biomass accumulation and fruit yield: OK

4. Conclusions: OK

**Acknowledgments: OK

**Author contribution: OK

**Competing interests: OK

**References: OK

Reviewer #2: 1. The author provided a detailed and sincere response to the reviewers' questions. However, given the article's overall design and results, it is not suitable for publication in this journal.

2. The litter of the silverbush has not undergone microbial fermentation and decomposition process. This makes its usage inconvenient. There are limitations in collecting raw materials, and transportation is also inconvenient. Therefore, this technology is difficult to apply on a large scale.

3. It is recommended that the author consider submitting the article to other widely read science or agriculture promotion journals.

.

Reviewer #1: **Yes:** Joao Paulo Saraiva MoraisJoao Paulo Saraiva MoraisJoao Paulo Saraiva MoraisJoao Paulo Saraiva Morais

Reviewer #2: **Yes:** Shifan YangShifan YangShifan YangShifan Yang

---

## [Author Response · Author response to Decision Letter 2]

13 Mar 2026

Response to Editor

◘ Editor Comment

Address the remaining points from Reviewer #1

Author Response

We appreciate your guidance. All remaining comments from Reviewer #1 have been carefully addressed in the revised manuscript accordingly. Please refer to the response to reviewer comments#1 given below and also see our revised manuscript.

◘ Editor Comment

Further edit the abstract by reducing the number of individual values and p-values. Emphasize the main effect sizes and trends.

Author Response

Thank you very much for your valuable comment. The Abstract has been revised to improve clarity and readability by reducing the number of individual numerical values and p-values. Instead of listing multiple statistical details, the revised abstract now emphasizes the overall effect sizes, major trends and key findings of the study. Our revised manuscript is now more improves in terms of narrative flow and highlights the primary outcomes of the research more clearly (please see L 28-29, 31-36). Revised abstract is now shown as:

“While the effects of different fertilization strategies on chili (Capsicum frutescens) cultivation have been previously examined, the lack of comparative assessment of mixed versus mono-fertilization approaches limits our understanding. To address this gap, we conducted an experiment using a completely randomized design (CRD) with five fertilization treatments along with four replications of each treatment (e.g., Leucaena leucocephala leaf litter and recommended dose of synthetic fertilizers (RDF)) to observed the effect of these treatments on productivity traits, NPP (net primary productivity), quality, and overall yield of chili. We found that mixed fertilization (T4) results higher yield (~ 60% and ~ 90%) compared to control (T0) and sole organic fertilizer (T1+T2) respectively. The T4 treatment showed consistently higher plant height, leaf and fruit number at 60 DAT, as well as higher flowering at 45 DAT compared to other treatments. We also found that NPP (above- and belowground combined) was higher in T3 and T₄ treatments compared to control and organic fertilization. Similarly, quality traits such as vitamin C, capsaicin content, and SPAD reading were higher under mixed fertilization. Linear fit regression model indicated that both ANPP and yield were positively associated with vegetative and reproductive traits, particularly leaf number and flowering, highlighting that structural growth directly contributed to productivity gains. Overall, our results suggest that mixed fertilization enhances both productivity and quality traits of chili. Therefore, integrated organic and inorganic fertilizer management are recommended to improve yield and quality of chili.”

◘ Editor Comment

Explicitly highlight in the Discussion and Conclusions that the fertilizer-treatment set is not a complete factorial of organic × inorganic levels and that this limit what can be inferred (mention what effects cannot be estimated). This is mentioned, but only once.

Author Response

Thank you for your comments. The Discussion and Conclusions sections have been revised and explicitly clarify that the fertilizer treatment structure does not represent a full factorial design of organic × inorganic fertilizer levels. As a result, the study cannot statistically estimate the independent main effects of organic and inorganic fertilizer treatment. The observed differences therefore represent comparisons among specific treatment combinations used in this study rather than a complete factorial analysis. As suggested, this limitation is now clearly stated in both the Discussion and Conclusions sections, and future research using a complete factorial design is also recommended (please see L 432-441and L 460-463).

◘ Editor Comment

Replace all speculative phrases (“we assume… may enhance”) with either a clear indication that these are hypotheses for future work or citations that support the proposed mechanisms.

Author Response

We apologize for using such speculative phrases. In the current form, we carefully reviewed the discussion section and replace all speculative phrases as suggested. Example of some revised sentences are:

Original text (speculative): “Although we did not measure the mechanism behind the findings, however, considering other experiment, we assumed that the presence of organic matter such as L. leucocephala leaf litter enhances rhizospheric activity and microbial interactions that release essential growth hormones like cytokinins and auxins.”

Revised text: ‘Previous studies have also reported that the presence of organic matter such as L. leucocephala leaf litter may enhance rhizospheric microbial activity and nutrient-mediated hormonal processes that influence flower retention and fruit development [13,14]. However, direct measurements of microbial dynamics and hormone regulation would be required to confirm these pathways’ (L 350-354). We also excluded those speculative words such as “could be due to, supposed to, this is because, although……. may enhances” in the revised text (please see L 369-372, 378-382, 384-388, 398-401).

Response to Reviewer

Response to Reviewer comments # 1

◘ Reviewer Comment

The document improved from the previous version, but the authors still must make corrections in the text. For example, the document needs an extensive English revision.

Author Response

We thank for your meaningful suggestion. The entire manuscript has now undergone a comprehensive English language revision using advanced grammar and style correction tools. In addition, all the co-authors of this manuscript carefully reviewed sentence structure, tense consistency, subject–verb agreement, clarity of expression, and scientific phrasing throughout the manuscript. We believe, the current revised version reflects substantial improvement in readability, fluency, and grammatical accuracy. Please refer to our manuscript.

◘ Reviewer Comment

There are no results or discussion for the whole section ‘2.2. Pot preparation and experimental design.

Author Response

Thank you for this observation. Section 2.2. Pot preparation and experimental design belong to the ‘Materials and Methods’ section and therefore we clearly described the details experimental procedures in the Materials and Methods rather than presenting results section. The results section substantially described our research findings.

However, to improve clarity and avoid possible confusion, we carefully revised this section to provide a clearer description of the experimental layout, pot preparation procedure, treatment structure, and replication used in the study. For example, see L 117-121, L 146-148. These revisions ensure that the experimental design is more clearly explained to readers.

◘ Reviewer Comment

The introduction contains many ideas within only two paragraphs. The reviewer suggests dividing the content into smaller paragraphs structured as follows:

• Importance of chili

• Importance of mineral nutrition with its shortcomings

• Importance of organic fertilizers with their shortcomings

• Advantages of combining both fertilizer types

• Objectives of the study

Author Response

Thank you for your helpful and structural suggestion. As suggested, the Introduction has now been reorganized and clearly defined paragraphs following exactly the recommended structure:

1. Importance of chili production (please see L 43-51)

2. Importance and limitations of mineral nutrition (L 52-57)

3. Role and limitations of organic fertilizers (L 58-64)

4. Rationale for integrated nutrient management (L 65-82)

5. Clear statement of study objectives (L 83-91)

◘ Reviewer Comment

Please explain the meaning of an abbreviation the first time it is written in the text. Therefore, please explain the meaning of TSP and MOP in line 124, not 149–150.

Author Response

Thank you for your comments. The abbreviations TSP and MOP are now fully defined at their first appearance in Section 2.2:

• TSP is defined as Triple Super Phosphate. (L 127)

• MOP is defined as Muriate of Potash. (L 127)

◘ Reviewer Comment

There is no result regarding the decomposition kinetics of the leucaena leaves. Please add this information to the document

Author Response

We appreciate this observation. The litterbag technique in this study was primarily used to ensure gradual nutrient release from Leucaena leucocephala leaf litter during the experimental. Now, we have revised and added a brief explanation of the decomposition dynamics in “2.2.2. Application of litterbag technique’’ section. We followed the negative exponential decay model to assess decomposition dynamics of L. leucocephala leaf litter (Olson, 1963). Please see L 173-177, 183-186.

◘ Reviewer Comment

Please pay attention to scientific names such as “L. leucocephala.” The species name must always be written in lowercase.

Author Response

We apologies for this issue. All scientific names have been carefully checked throughout the manuscript as advised. The genus and species name Leucaena leucocephala is now consistently written following standard scientific nomenclature, with the genus capitalized and the species epithet in lowercase.

---

## [Editor Report · Decision Letter 2]

16 Mar 2026

Beyond mono fertilization: Mixed fertilization enhances productivity and quality of chili (Capsicum frutescens)

PONE-D-25-44796R2

Dear Dr. Jaman,

We’re pleased to inform you that your manuscript has been judged scientifically suitable for publication and will be formally accepted for publication once it meets all outstanding technical requirements.

Kind regards,

Randall P. Niedz

Academic Editor

PLOS One
---

## [Editor Report · Acceptance letter]

PONE-D-25-44796R2

PLOS One

Dear Dr. Jaman,

I'm pleased to inform you that your manuscript has been deemed suitable for publication in PLOS One. Congratulations! Your manuscript is now being handed over to our production team.

Kind regards,

on behalf of

Dr. Randall P. Niedz

Academic Editor

PLOS One